# Peptide Shuttles for Blood–Brain Barrier Drug Delivery

**DOI:** 10.3390/pharmaceutics14091874

**Published:** 2022-09-05

**Authors:** Macarena Sánchez-Navarro, Ernest Giralt

**Affiliations:** 1Department of Molecular Biology, Instituto de Parasitología y Biomedicina ‘‘López Neyra” (CSIC), 18016 Granada, Spain; 2Institute for Research in Biomedicine (IRB Barcelona), Barcelona Institute of Science and Technology (BIST), Baldiri Reixac 10, 08028 Barcelona, Spain; 3Department of Inorganic and Organic Chemistry, University of Barcelona, Martí i Franquès 1-11, 08028 Barcelona, Spain

**Keywords:** blood–brain barrier, BBB–peptide shuttle, brain delivery

## Abstract

The blood–brain barrier (BBB) limits the delivery of therapeutics to the brain but also represents the main gate for nutrient entrance. Targeting the natural transport mechanisms of the BBB offers an attractive route for brain drug delivery. Peptide shuttles are able to use these mechanisms to increase the transport of compounds that cannot cross the BBB unaided. As peptides are a group of biomolecules with unique physicochemical and structural properties, the field of peptide shuttles has substantially evolved in the last few years. In this review, we analyze the main classifications of BBB–peptide shuttles and the leading sources used to discover them.

## 1. Introduction

Neurological-related disorders, such as glioblastoma, and other central nervous system (CNS) tumors, Parkinson’s disease (PD), migraine or stroke, to name a few, are a major cause of death and disability worldwide, being the second most common cause of death after cardiovascular diseases [1]. Despite the efforts to treat those diseases and to improve the quality of lives of patients and families, there are not yet any efficient treatments available. The main limitation in the development of such treatments is the presence of the blood–brain barrier (BBB) [2]. The BBB isolates and protects the brain from harmful blood-borne substances and is the main gate for nutrient entrance [3].

The early idea of a BBB shuttle was developed by Pardridge, who thought the natural transport mechanism of certain peptides and proteins could be explored to deliver pharmaceutics to the brain [4]. Since then, several types of compounds able to hijack the natural transport mechanism at the BBB have been developed [5,6,7,8]. Monoclonal antibodies targeting the receptors and transporters present at the BBB have been established to deliver a wide variety of biotherapeutics [9,10,11,12]. For instance, a monoclonal antibody against the human insulin receptor conjugated to the enzyme α-L-iduronidase is being evaluated in a clinical trial (NCT03053089 and NCT03071341) to treat mucopolysaccharidosis I [13]. However, antibodies display a very high affinity for their targets, which may hamper the dissociation from the receptor, leading to vesicle entrapment and inefficient transport across the BBB [10].

In 1999, Schwarce et al. proved that a cell-penetrating peptide, TAT, delivered an active enzyme to the brain parenchyma [14]. The delivery was not selective, but this result opened a field of investigation. Since then, more than 40 peptides have been described as able to cross the BBB carrying compounds that cannot transverse this membrane alone [8,15]. In this paper, we discuss the main families of BBB–peptide shuttle as well as the most explored sources to discover them.

## 2. The Blood–Brain Barrier

The presence of the BBB ensures brain homeostasis. This highly metabolic and physical barrier allows the passage of only a selected group of nutrients, such as sugars or amino acids, restricting the entrance of harmful substances. The natural transport mechanisms present at the BBB that tightly control the access of nutrients to the brain can be divided into passive and active according to their energy requirements (Figure 1). Passive transport mechanisms embrace transcellular passive diffusion and paracellular diffusion, while active transport mechanisms include receptor- and transporter-mediated transcytosis and adsorptive-mediated transcytosis. The main physiological characteristic of the BBB is that its constitutive endothelial cells are tightly bound by the presence of tight and adherens junction proteins that limit the paracellular transport of substances. In addition, this barrier has reduced vesicular transport, bearing high proteolytic activity and presenting efflux pumps at its abluminal side that force the exit of potentially toxic substances. However, the high vascularization of the brain offers a unique platform for the delivery of therapeutics [16]. Each neuron has a capillary of less than 20 µm [17]. If a compound is able to undergo transcytosis after interacting with a receptor at the BBB, it would be homogenously distributed along the brain. This unique feature has prompted the study of several types of ligands targeting receptors at the luminal side of the BBB. Antibodies against the transferrin (Tf), insulin, or low-density protein-1 (LRP1) receptors have been widely studied with varying degrees of success [13,14,15,16,17,18]. 

In order to exploit the natural transport mechanisms present at the BBB for the delivery of therapeutics, several BBB peptide shuttles, able to increase the transport of compounds of interest, have been developed [5,6,7,8]. On the one hand, small lipophilic peptides, such as diketopiperazines (DKPs) [19], N-methyl phenylalanines [20], or phenyl prolines [21] have been evaluated as carriers of small drugs by targeting passive transport mechanisms. On the other hand, cell-penetrating peptides or peptides targeting receptors have been proposed for the brain delivery of drugs by targeting active transport mechanisms.

## 3. Peptides Designed to Increase Passive Transport of Drugs

Transcellular and paracellular passive diffusion has traditionally been envisaged as a strategy for the delivery of small lipophilic compounds. The use of small peptides that cross by this mechanism has been explored for the delivery of small drugs. One of the most representative examples is the use of diketopiperazines (DKPs) [19], highly stable cyclic dipeptides that are able to increase the transport of two small compounds with therapeutic interest, L-dopa and baicalin, by means of passive diffusion, as proved by the PAMPA assay, the gold standard model used to evaluate this mechanism. In addition, this family of compounds was used to deliver a hexapeptide, able to inhibit Tau aggregation in vivo in mice [22].

Passive diffusion passage is governed by the physicochemical properties of the compounds. For instance, the number of hydrogen acceptors and donors is an important parameter to optimize. In general, the higher the lipophilicity, the higher the transport, but if a compound is too hydrophobic, it can be retained at the lipid membrane. Modifications such as *N*-methylation [23] or halogenation [20,24] have been used as powerful tools to modulate the lipophilicity of small peptides. To this end, a family of BBB shuttle peptides composed of *N*-methylated tetrapeptides was proposed as efficient vectors to increase the transport of small drugs such as L-dopa [23]. Importantly, this chemical modification also increases the stability of the peptide shuttle to serum proteases. The potential *N*-methyl phenylalanines based shuttles have been explored through modifications to its structure with amino acids with different stereochemistry and a different number of halogenated atoms, yielding optimized peptide shuttles with an ideal structure for given cargoes [20]. For instance, when comparing how di-peptides-based *N*-methyl phenylalanines increase the transport of 3,4-dihydroxy-l-phenylalanine, 4-aminobutyric acid, and nipecotic acid as cargoes, the authors found that the first, which is more polar, was better transported by *N*-methylated peptide shuttles, while the last two were better transported by chlorinated-*N*-methylated peptide shuttles [20]. This work suggested that slight modifications of the structural properties of a given peptide shuttle for a given cargo can lead to optimized transport. The main drawback of this family of compounds is their limited solubility, which can be overcome using phenylproline-based peptides [21].

Another strategy, which has lately been gaining attention, is the use of peptides able to increase the porosity of the tight junctions, thus enabling the delivery of compounds. To this end, peptides derived from claudin-5 [25], E-cadherin [26], or occluding [27], which have been proved to interact with the proteins that form the tight junctions, are able to modulate the protein-protein interactions that hold these protein connections. For instance, HAV6 (Ac-SHAVSS-NH2) derived from the C-1 domain of E-cadherin, is able to increase the paracellular transport of anticancer drugs, magnetic resonance imaging (MRI) contrast agents, or near-infrared dyes [26,27,28]. More recently, this peptide has been compared with ADT5 (Ac-C(&)DTPPVC(&)-NH2) [29], another E-cadherin-derived peptide, for the delivery of proteins [28]. Lysozyme, albumin, IgG mAb, and fibronectin with 15, 65, 150 and 220 KDa, respectively, were intravenously co-administered in mice. ADT5 increased the transport of lysozyme, albumin, and an IgG mAb but not fibronectin, while HAV6 only improved the delivery of lysozyme [28]. Mechanistic studies revealed that these peptides are able to promote the formation of pores within the protein tight junctions of enough size to allow the transport of proteins. The authors proposed that the formed pores are of different sizes, with the largest being the least stable. Thus, depending on the duration of the effect caused by the peptide modulator, the transport of big proteins will be limited [28]. This work demonstrated that tight junction modulation can be used for the delivery of therapeutic proteins, although important factors such as the size of the protein to delivery must be considered. It might be necessary to adapt the selected tight junction modulator to the size of the protein cargo to avoid the passage of bigger proteins that could have undesired effects [29]. Despite the potential of this strategy, significant concerns about its safety must be considered. Precise control of the duration of the effect, to limit the passage of toxic substances, for instance, is of utmost importance.

Other families of peptides, such as membrane-active peptides [30,31], are able to promote the transient opening of the BBB. Melittin, a venom-derived peptide, was recently shown to promote reversible BBB opening during 24 h at neurologically safe sub-toxic concentrations [31].

## 4. Peptides Designed to Increase Active Transport of Drugs

Targeting the active transport mechanisms of the BBB represents another attractive option for brain delivery. However, the search for an efficient receptor for the delivery of therapeutics directed to the CNS is challenging. Such a receptor should be expressed at higher levels in the brain microvasculature than in the peripheral tissues or in the brain parenchyma. As such, the risk of off-target effects would be minimized. In addition, the receptor must be able to undergo transcytosis at a reasonable rate, allowing for the passage of the selected cargo from the blood to the brain. To identify such a receptor, several proteomic and transcriptomic works have been carried out [31,32,33]. Importantly, these studies allow for comparison of the level of expression of a given receptor between preclinical species, such as mice or rats, and humans because the difference in expression patterns can hamper the development of delivery agents [33]. As an example, TfR, which has been widely used as a model receptor for brain delivery, is expressed five-fold higher in mice brain microvasculature than in that of humans [34]. Another important fact to consider is that the sole enrichment of the mRNA of a receptor at the brain microvasculature does not make it suitable as a target for brain delivery. For instance, Tam et al. identified Ldlrad3 and CD320 [35] as possible targets for brain delivery, but evaluation of the transport of monoclonal antibodies against these two receptors indicated that they did not have preferred brain uptake, showing similar levels to control IgG [34].

Several peptides have been shown to increase the transport of drugs by targeting the active transport mechanism of the BBB, mainly through targeting the low-density lipoprotein (LDL) and transferrin receptors [8,15]. For instance, Angiopep-2, which was proven to interact with the LRP-1 [36], has been used to modify nanoparticles of different nature [37,38], peptides [39,40], proteins [41,42], and small molecules [43,44], increasing their transport in several in vivo and in vitro models. Another example is THR, which was discovered through phage display against cells overexpressing the human transferrin receptor [45], and it was shown to deliver gold nanoparticles to the brain parenchyma of mice [8].

## 5. Sources of BBB Shuttles

### 5.1. Natural Proteins

Natural proteins have served as an inspiration for developing new brain-targeting peptides (Table 1). For instance, peptides based on apolipoproteins have been widely explored. Apolipoproteins are involved in lipid and cholesterol trafficking and interact with the LDLRs that are present at the BBB. Peptides based on ApoE and ApoB proteins have been used to modify various enzymes to develop new therapies for enzyme replacement therapy [46,47,48]. The most successful example of a BBB shuttle inspired by natural sources is angiopep-2, a 19-mer peptide derived from the alignment of the Kunitz domain of human proteins that interact with the LRP-1 [35]. Remarkably, angiopep-2 modified with three molecules of paclitaxel (ANG1005) has been evaluated in various clinical trials, showing good safety, tolerability, pharmacokinetics, and efficacy in patients with advanced solid tumors (NCT02048059), high-grade glioma (NCT01967810) [49,50], and leptomeningeal carcinomatosis and brain metastasis from breast cancer (NCT01480583 and NCT02048059) [50]. In the near future, a new trial (NCT03613181) will evaluate the effect of ANG1005 in HER2-negative breast cancer patients with the newly diagnosed leptomeningeal disease and previously treated brain metastases (source: www.clinicaltrials.gov, accessed on 1 July 2022).

Melanotransferrin (MTf), or p97, is an 80 kDa protein able to bind iron to transport it across the central nervous system [51]. Its soluble form was shown to undergo transcytosis across the BBB [52]. The potential of this protein as a shuttle has been explored in the transport of small molecules or antibodies [53,54]. A 12-mer peptide derived from MTf was described upon evaluation of the tryptic mixture of this protein in a BBB cell-based model [55]. The selected peptide, DSSHAFTLDELR, preserves the capacity of MTf for crossing the BBB and is found in neurons, astrocytes, and microglia.

The rabies virus has clear CNS tropism [61]. The protein responsible for virus internalization is the trimeric glycoprotein known as RVG, which was shown to interact with the α subunit of AchR. Lentz et al. [62] compared the sequence of RVG with some snake venom toxins that interact with AchR and defined the region between amino acids 175 and 203 of the RVG protein as the most efficient for binding. The peptide RVG29 comprises a nonimmunogenic region of the RVG protein, which made it interesting as a BBB-shuttle peptide. Since the first seminal work where an RVG29 nanosystem was used to deliver small-interfering RNA in vivo in mice [57], several researchers have explored the use of this peptide [63]. Most of the studies modified RVG29 with several arginines in order to enhance the complexation of nucleic acids. The use of this extension may alter the internalization mechanism of the RVG29 peptide.

The dengue virus capsid protein has served as a scaffold for the design of various BBB peptide shuttles [58], which have recently been used to modify an Fc domain of IgG without affecting its binding properties to the FcR [64] or to modify porphyrins to yield peptide−porphyrin conjugates that can be used as antiviral drugs [65].

Venoms are a privileged source of bioactive compounds [66,67]. One of their major components is peptides of complex structure, which are characterized by a rich content on disulfide bridges that confer them high metabolic stability. Currently, there are 11 compounds derived from venoms approved by the FDA [67]. Some venoms affect the CNS, serving as motivation for researchers to search for CNS active compounds. To this end, two BBB-shuttle peptides derived from venoms have been described: MiniAp-4 and miniCTX3 [59,60]. MiniAp-4 is a minimized version of apamin, which is the main component of bee venom. This highly stable peptidomimetic was shown to be able to deliver a fluorophore in vivo in mice [59]. MiniCTX3 corresponds to a minimized version of chlorotoxin [60], a disulfide-rich peptide from the venom of the Israeli scorpion *Leiurus quinquestriatus* [68]. MiniCTX3-modified gold nanoparticles translocated across a human-cell-based BBB model.

### 5.2. Phage Display

Phage display is a potent source of bioactive peptides that have been widely used to obtain BBB-shuttle peptide candidates [69]. It consists of the evaluation against a target of interest of a library of bacteriophages, where each one is genetically modified to display a given peptide or protein on its coat protein. Phage display libraries have been evaluated against isolated receptors or proteins [70,71,72], cellular models [45,73,74,75], an even living animals [76,77,78,79,80,81,82,83,84,85,86,87]. A list of BBB shuttle peptides discovered by phage display is summarized in Table 2.

The validation of the selected targets is of great importance because several factors, such as the affinity for albumin or plastic, can affect the replication of a given phage, biasing the results. To this end, several target-unrelated peptides (TUPs) have been described [69]. For instance, HAIYPRH, which was initially discovered as a transferrin receptor binder [45], has been found in more than 30 phage display experiments against more than 20 different targets (source: Biopanning Data Bank (BDB) [88]). A combination of in vitro and in vivo screening methods is suggested as a way to minimize the discovery of TUPs [69]. However, some peptides can interact in a nonspecific manner with various receptors or cell membrane components and can be useful for more than one application, although their promiscuity would need to be assessed in each case.

### 5.3. Chemical Libraries

High-throughput screening techniques have sped up the development of therapeutics during the last few decades [89]. In the field of peptide drug discovery, phage display is the main example, although it has some limitations, such as the restricted possibilities for including nonproteinogenic amino acids. The use of DNA- or mRNA-encoded libraries [90,91] or one-bead-one compound (OBOC) [92] libraries overcome this restriction. This last technology has been used to discover new protease-resistant BBB-shuttle peptides. Guixer et al. [93] evaluated for the first time an all-D OBOC library against a BBB cell-based model. Detection of the peptides able to transverse the cell monolayer was performed by mass spectrometry analysis. More recently [94], an OBOC library also composed of D amino acids was used to discover brevican-targeting peptides. Brevican is an extracellular matrix protein located in the CNS and overexpressed in glioma cells. One of the peptides discovered was found to cross the BBB in vivo in mice [94] and to shuttle the insoluble drug camptothecin in an orthotopic mice model [95].

### 5.4. Optimization

Protease liability is one of the major concerns in the development of therapeutic peptides. To overcome this limitation, several strategies have been applied, such as the use of non-natural amino acids, cyclisation, or chemical modifications [96]. In the field of BBB shuttles, one of the most common methods is the use of the retro-enantio or retro-inverso approach, which consists of the preparation of a given peptide using D amino acids and reversing the order of the sequence. To this end, the topological properties of the parent peptide and its retro-enantio counterpart will be very similar (Figure 2). Thus, this strategy has been applied to angiopep-2 [97], THR [98], CDX [99], and to a minimized version of RVG [100]. In all the cases, the newly designed peptides displayed higher stability to serum proteases and proved to be more efficient in the transport of various cargoes across the BBB, both in vitro and in vivo. In addition, it was demonstrated that retro-enantio/inverso peptides are less immunogenic than the original peptides, making them very attractive for their development as therapeutics [101]. For instance, THRre was recently used to efficiently deliver amphiphilic polymeric nanoparticles loaded with a cytotoxic drug in a diffuse intrinsic pontine glioma model [102].

### 5.5. Computational Prediction

The use of computational methods to predict the BBB permeability of peptides is very attractive due to their low cost and the rapid evolution of the field [103]. These methods use chemoinformatic filters, molecular dynamic simulations, statistical models, and/or artificial intelligence algorithms [103]. Pioneering work was conducted by Giralt et al., who worked on the design of genetic algorithms to decipher the key features necessary for a peptide to cross the BBB [104,105]. Since then, several predictors have been developed, such as the sequence-based predictor BBPpred [106]; B3predict, which is based on machine-learning models [107]; and the online tool BBPpredict [108]. In addition, databases such as B3Pdb [109] and Brainpeps [110] hold relevant information about already described BBB-shuttle peptides.

Despite the increasing number of predictors and the impressive evolution of the field, there are still a few issues that remain to be resolved. The prediction of BBB permeability based on the physicochemical properties, such as topological polar surface area or the number of hydrogen bond acceptors, among others, only considers passive transport mechanisms across the lipid bilayer, neglecting the active transport mechanisms such as a receptor- or adsorptive-mediated transcytosis. Additionally, machine learning models are limited by the amount and quality of the currently available data [103].

More sophisticated methods are needed to find good BBB-shuttle peptides by computational prediction. For instance, molecular dynamic simulations are time-consuming, both in computing and processing time [103]. Current efforts to overcome these limitations are directed to reduce the computational cost by implementing more realistic membrane compositions, which will also allow for the evaluation of different species and new sampling techniques [111,112].

## 6. Conclusions

The development of peptides as therapeutic entities is in a golden era [113,114]. Peptides have great properties, such as low immunogenicity and biocompatibility. In addition, the advances in synthetic methodologies as well as in the strategies to increase their circulation time and stability to proteases have helped to overcome their main drawbacks. As a consequence, there are more than 150 peptides in clinical development [115]. In this context, the field of peptide shuttles for brain delivery has notably evolved, with a few candidates in clinical development.

Despite the evolution of the field, a better understanding of BBB properties and composition is needed to develop new and more efficient BBB-shuttle peptides. In this way, the discovery of new receptors may be accomplished by studies based on proteomic and transcriptomic approaches [32,33,34]. In addition, strategies to efficiently characterize the different transport mechanisms undertaken by a given peptide shuttle have to be further developed. Then, nonbiased strategies to discover BBB-shuttle peptides, such as in vivo phage display or in vivo screening of synthetic libraries, can be applied to the discovery of new peptides without missing information about the mechanism used.

## Figures and Tables

**Figure 1 pharmaceutics-14-01874-f001:**
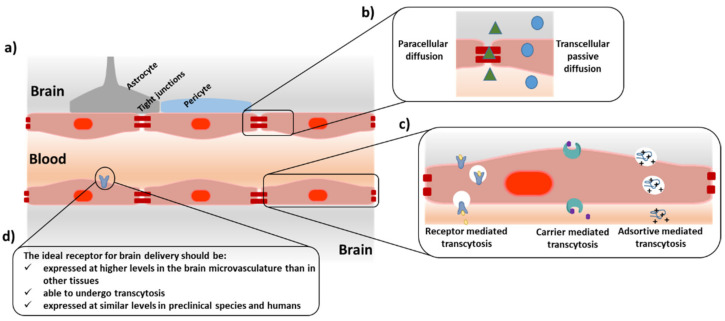
Schematic representation of the blood–brain barrier structure (BBB). (**a**) The blood–brain barrier comprises a monolayer of endothelial cells in intimate contact with astrocytes’ end-feet and pericytes. The endothelial cells are strongly bound by tight junction (TJs) proteins. (**b**) Passive transport mechanisms are divided into paracellular diffusion and transcellular passive diffusion; (**c**) active transport mechanisms include transcytosis mediated by receptors and transporters and adsorptive-mediated; (**d**) minimal requirements of a receptor for targeted brain delivery.

**Figure 2 pharmaceutics-14-01874-f002:**
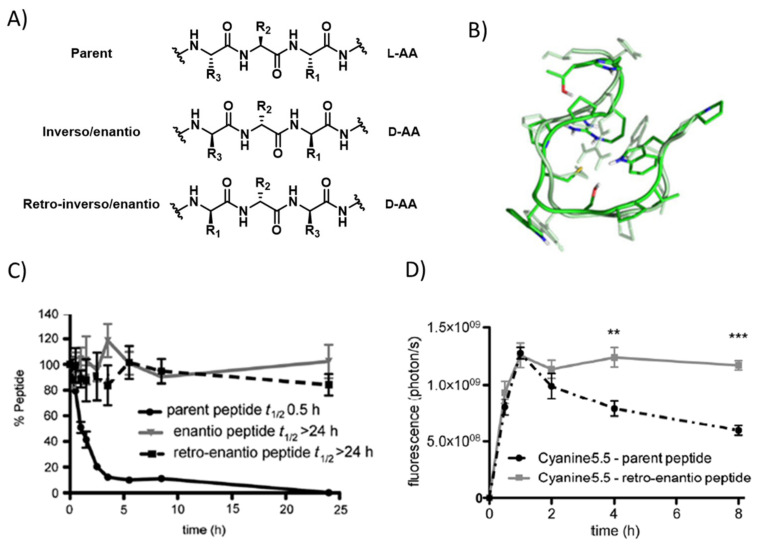
(**A**) Structure of a natural peptide and its retro and retro-enantio/inverso counterparts; (**B**) three-dimensional superposition of one pairing obtained from the cross-RMSD matrix of THR and THR retro-enantio [101]; (**C**) percentage of peptide versus incubation time in human serum obtained for THR and its protease-resistant enantio, and retro-enantio analogues [98]; (**D**) in vivo fluorescence quantification measured in a preclinical IVIS spectrum in vivo imaging system (IVIS-200) at 0.5, 1, 2, 4, and 8 h after injection of cyanine 5,5-THR and cyanine 5,5-THR retro-enantio [98] Error bars correspond to standard error mean (s.e.m.). Unpaired t student test: ** *p* < 0.01, *** *p* < 0.001.

**Table 1 pharmaceutics-14-01874-t001:** Brain-targeting peptides obtained from natural sources.

Peptide	Origin	Target	Ref
(LRKLRKLL)_2_	ApoE (Aa 141–149)_2_	LDLR	[46,47]
TEELRVRLASHLRKLRKRLLRDA	ApoE (Aa 130–152)	LDLR	[47]
SVIDALQYKLEGTTRLTRKRGLKLATALSLSNKFVEGS	ApoB (Aa 3371–3409)	LDLR	[47]
TFFYGGSRGKRNNFKTEEY	Sequence alignment of human Kunitz domains	LRP1	[36]
DSSHAFTLDELR	MTf (Aa 441–452)	LDLR	[56]
YTIWMPENPRPGTPCDIFTNSRGKRASNG	RVG glycoprotein (Aa 175–203)	AchR	[57]
VQQLTKRFSL	DEN2C ^ [a]^ (Aa 26–35)	None	[58]
KLFMALVAFLRFLT	DEN2C (Aa 45–59)	None	[58]
AGILKRW	DEN2C (Aa 63–69)	None	[58]
KSKAINVLRGFRKEIGRMLNILN	DEN2C (Aa 74–97)	None	[58]
[Dap](&)KAPETALD(&) ^ [b]^	Apamin	Unknown	[59]
[Dap](&)YGPQD(&)	Chlorotoxin	Unknown	[60]

^ [a]^ DEN2C: Dengue virus type 2 capsid protein. ^ [b]^ [Dap] is the three-letter code for L-2,3-diaminopropionic acid; (&) refers to cyclic peptides [8].

**Table 2 pharmaceutics-14-01874-t002:** Brain -targeting peptides discovered by phage display.

Peptide	Target	Panned Against	Ref
C(&)LSSRLDAC(&)	Brain	BALB/c mice	[76]
GHKAKGPRK	hTfR	hTfR	[70]
THRPPMWSPVWP	TfR	hTfR (chicken fibroblast)	[45]
HLNILSTLWKYR	GM1	Trisialoganglioside (GT1b)	[71]
C(&)AGALC(&)Y	Brain endothelium	BALB/c, FVB/N, and C57BL mice	[77]
GLAHSFSDFARDFV	Brain endothelium	C57Bl/6 and BALB/c mice	[80]
GYRPVHNIRGHWAPG	Brain endothelium	C57Bl/6 and BALB/c mice	[80]
TGNYKALHPHNG	Brain	ICR mice	[81]
C(&)RTIGPSVC(&)	Apo-TfR	BALB/c mice	[82]
C(&)TSTSAPYC(&)	Brain	ICR mice	[83]
C(&)SYTSSTMC(&)	Brain	Sprague-Dawley rats	[84]
DSGLC(&)MPRLRGC(&)DPR	LDLR	hLDLR	[72]
TPSYDTYAAELR	Brain through the BCSFB	Sprague-Dawley rats	[85]
RLSSVDSDLSGC	BBB/BCSFB	Wistar rats	[86]
SGVYKVAYDWQH	Brain endothelium	Human BBB cellular model	[73]
TFYGGRPKRNNFLRGIRSRGD	BBB/BTB	BALB/c mice	[87]
C(&)SLSHSPQC(&)	Brain endothelium	hCMEC/D3 cell monolayers	[74]
VAARTGEIYVPW	Brain endothelium	Primary endothelial rat cellular model	[75]
GLHTSATNLYLH	Brain endothelium	Primary endothelial rat cellular model	[75]
C(&)SLSHSPQC(&)	Brain endothelium	hCMEC/D3 cell monolayers	[74]
C(&)RGGKRSSC(&)	CNS	Ex vivo and in vivo EAE ^ [a]^ mice	[79]
QFAALPVRAHYG	Brain	C57BL/6J mice	[78]

^ [a]^ EAE: experimental autoimmune encephalomyelitis.

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
