# Peer review of "Peptide Shuttles for Blood–Brain Barrier Drug Delivery"

_pharmaceutics, 2022, doi:10.3390/pharmaceutics14091874_

Round 1
Reviewer 1 Report
The authors in the present article summarized BBB-peptides shuttles used for the delivery of therapeutics to the brain. The authors in detail described different types of shuttle peptides analyzed by in vitro studies or used in clinical trials. The article is clearly written with references to the latest articles. I recommend accepting the article in its present form.Author Response
We thank the reviewer for their nice revision.
Reviewer 2 Report
Giralt et al. provide a very nice summary of peptide shuttles utilized in blood-brain barrier delivery in this review paper. Representative examples reported over the past few years are thoroughly included. This manuscript is well constructed, and brief but sufficient background information was introduced. Peptides exhibit great potential in the drug delivery field, proven by their therapeutic efficacy. Thus, this manuscript fits the scope of Pharmaceutics, and I would recommend the publication in its current form.
Author Response
We thank the reviwer for their positive review.
Reviewer 3 Report
This review provides a nice update on the recent advancements in the area of utilizing peptide shuttles for BBB delivery. It largely focuses on peptides designed for improving passive diffusion or active transport, with limited discussion of cell penetrating peptides. Phage display is particularly highlighted as a potential tool for further identification of peptides. It parallels many similar recent reviews, while adding some additional insight in some areas. The manuscript suffers from poor use of the English language that significantly hampers readability.
Comments
Unusual phrases and grammatical decisions significantly impair understanding of this document. They are most concentrated in the introduction and conclusions sections. A few examples (not a comprehensive list):
The use of the word guarantee in line 48
The use of the word impaired in 58
“compounds of different nature” line 32
This leads readers to the incorrect conclusion that the authors are not knowledgeable in the subject area, despite the technical expertise shown in the body of the document.
The authors comment on the limitation of the size of cargo that can be carried through by tight junction modifying peptides. It would be beneficial if they could expand on this idea more directly following this statement.
The authors should include this recent work describing the use of melittin for transient BBB opening. https://doi.org/10.1016/j.biomaterials.2021.120942
The final paragraph on computational modeling fails to actually make and conclusions. What are the current limitations? What efforts have been made to overcome these?
Why is targeting active transport mechanisms the most attractive? Line 124
The authors have made a very nice table 1 for peptides identified through phage display screens, could similar tables be made for the additional sections? This also seems to be the most well citated section
Lines 24-6 – I agree that there are few treatments available for many neurological related disorders, but claiming that there are “no current efficient treatments” for the entire class of disorders is a stretch.
Lines 99-102 are difficult to follow – more details about the cargo would also be helpful.
The sentence that stretched 114-116 needs a citation. Similarly the one that stretches 131-134 also requires a citation.

Author Response
Reviewer 3 is acknowledged for their insightful revision. Their comments are addressed in the attached document.

Round 2
Reviewer 3 Report
This is a well written review that nicely covers the recent updates in the areas of peptide shuttles for BBB delivery. The authors addressed all of the major concerns and improved the overall readability of the document.